# Overexpression of *GhCAD6* in Upland Cotton (*Gossypium hirsutum* L.) Enhances Fiber Quality and Increases Lignin Content in Fibers

**DOI:** 10.3390/ijms26199518

**Published:** 2025-09-29

**Authors:** Zumuremu Tuerxun, Chenyu Li, Xiaorong Li, Yuanxin Li, Xinxin Qin, Hui Zhang, Yang Yang, Guo Chen, Juan Li, Zhigang Liu, Xunji Chen, Darun Cai, Bo Li

**Affiliations:** 1Xinjiang Uygur Autonomous Region Academy of Agricultural Sciences Biological Breeding Laboratory/Xinjiang Key Laboratory of Crop Biotechnology, Urumqi 830091, China; azze128@163.com (Z.T.); lcy2023123666@163.com (C.L.);; 2College of Life Sciences, Xinjiang Normal University, Urumqi 830054, China

**Keywords:** cotton, overexpression, *GhCAD6*, lignin, fiber quality

## Abstract

Cotton is a vital economic crop, and cotton fiber serves as the primary raw material for the textile industry. Lignin in cotton fiber is closely associated with fiber quality. Lignin is synthesized through the phenylpropanoid metabolic pathway, where the cinnamyl alcohol dehydrogenase gene *CAD6* plays a significant role. In this study, we obtained successfully transformed overexpression plants by constructing an overexpression vector and performing genetic transformation and tissue culture. To verify the function of the *GhCAD6* gene in upland cotton, we analyzed the agronomic traits, fiber quality, cell wall structure, and lignin content of *GhCAD6*-overexpressing plants. Our results indicate that the *GhCAD6* gene is predominantly expressed during the stages of fiber elongation and secondary wall synthesis. Overexpression of the *GhCAD6* gene resulted in increased plant lignin content and fiber upper half mean length, boll number per plant, fiber uniformity index, strength, and lint were improved. The fiber surface was smoother, and the fiber cell wall was more compact. These findings demonstrate that the *GhCAD6* gene positively regulates lignin synthesis and fiber quality formation, contributing to the enhancement of cotton fiber quality.

## 1. Introduction

Cotton (*Gossypium* spp. L.) is a significant economic crop. It has the highest purity among natural fibers and holds substantial value in the textile industry [1]. Cotton fibers are the longest and fastest-growing single cells among plant cells [2], originating from the differentiation and development of ovule outer epidermal cells. Its development progresses through four stages: differentiation, elongation, thickening, and dehydration maturation. Each of these stages uniquely impacts the development of cotton fibers [3,4,5]. Mature cotton fiber cells typically reach lengths of 3 to 5 cm and are primarily composed of cellulose, along with minor amounts of non-cellulosic carbohydrates such as pectic polysaccharides, xylan, lignin, and lignin-like phenols. Therefore, cotton fiber serves as an ideal model for studying plant cell elongation and cell wall genesis [6,7]; Previously, research has primarily focused on the genes involved in the biosynthesis of cellulose, pectin, and pectin precursors [8,9,10,11]. However, there has been limited investigation into the synthesis of lignin or lignin-like phenolic compounds.

Lignin is a crucial phenolic compound in plants, second only to cellulose. It is a key component of the intact cell walls in most vascular plants [12,13]. It was found that lignin biosynthesis is divided into three parts, namely, mustard alcohol, coniferol, and coumarol were synthesized by the shikimic acid pathway and phenylpropane pathway, and then, through the specific pathway of lignin synthesis and a series of complex biochemical reactions, the monomers were finally connected by different chemical bonds. Lignin is present in the secondary walls of plant cells, enhancing the overall mechanical strength of plants, facilitating the long-distance transport of water and other vital substances, and bolstering plants’ resistance to biotic and abiotic stresses [14,15,16]. Previously, it was believed that lignin deposition did not occur in the secondary walls of cotton fiber cells [17]. However, with the deepening of research, this view has been overturned. In cotton, lignin and lignin-like phenols are widely distributed in stems, roots, and leaves, reflecting their roles in various tissues and cells [18,19]. When cotton is invaded by pathogens, it activates the expression of lignin-synthesis-related genes and increases the activities of phenylalanine ammonia-lyase and peroxidase, leading to lignin deposition and enhanced resistance to pathogens [20]. More importantly, lignin and lignin-like phenols have a decisive effect on fiber quality [21]. With the advancement of textile technology, the demand for high-quality cotton fiber continues to increase. Therefore, the biosynthesis of lignin in plants can be regulated through various modern biotechnological methods, allowing for adjustments in lignin composition and content as needed. This regulation is of great significance for improving cotton fiber quality.

Cinnamyl alcohol dehydrogenase (CAD) is one of the earliest studied enzymes in lignin synthesis, playing a crucial role in this process [22]. The final step involving CAD in the lignin biosynthesis pathway contributes to the formation of guaiacyl lignin (G-lignin), syringyl lignin (S-lignin), and hydroxyphenyl lignin (H-lignin) [23,24]. *CAD* genes and their homologs have been identified in various plant species. These *CAD* genes are present as members of multiple gene families within plants. Arabidopsis (*Arabidopsis thaliana* L.) has 17 *CAD-like* genes [25]. In tobacco (*Nicotiana tabacum* L.) 10 *CAD* genes were identified [26]. Upland cotton (*Gossypium hirsutum* L.) contains 29 *CAD* genes [27], while mulberry (*Morus alba* L.) has 5 *CAD* genes [28]. Additionally, there are 25 *CAD* genes in pomegranate [29]. Several studies have indicated that the *CAD* gene is associated with lignin synthesis and is predominantly expressed during the formation of secondary walls in fiber development. Furthermore, double mutations in the *CAD4* and *CAD5* genes of *Arabidopsis thaliana* resulted in stems that were softer and more prone to inversion compared to wild-type plants. This mutation also led to a significant reduction in vascular tissue and a 94% decrease in lignin content, causing the entire plant to invert [30]. Down-regulating the expression of the *CAD* gene in tobacco, it was found that the stem cell walls of the *CAD* gene down-regulated plants were thinner, the vessel element density was lower, and the number of secondary xylem cells was reduced compared to wild-type plants [31]. Overexpression of the wild soybean (*Glycine max* (L.) Merr.) *GsCAD1* gene enhances resistance to soybean mosaic virus [32]. Similarly, overexpression of the *PpCAD1* gene increases resistance to *Botrytis cinerea*, and its heterologous expression in *Arabidopsis thaliana* also enhances resistance to *Botrytis cinerea* infection [33].

In this study, we conducted a preliminary investigation into the function of the *GhCAD6* gene and discovered that *GhCAD6* is predominantly expressed during the secondary wall thickening stage of fiber development. Overexpression of *GhCAD6* resulted in increased lignin content and improved fiber quality. These findings facilitate a deeper understanding of the *GhCAD6* gene’s function and the potential enhancement of fiber quality through genetic manipulation.

## 2. Results

### 2.1. Cloning of GhCAD6 Gene and Identification of Overexpression Plants

To investigate the role of *GhCAD6* in lignin synthesis, we constructed the overexpression vector pCAMBIA 2300-E6-*GhCAD6*, utilizing the E6 fiber-specific promoter (Figure 1a). This vector was transformed into the upland cotton variety Zao 36 via, resulting in the generation of *GhCAD6* overexpression plants. Fourteen positive plants were identified using two sets of specific primers: NPT II-F/R and E6-F/*GhCAD6*-R (Appendix A). Additionally, single-copy insertion was confirmed through Southern blot analysis. The results indicated that 14 positive clones were obtained (Figure 1b), and transgenic progenies from four single-copy lines were detected for further study (Figure 1c).

### 2.2. Expression Analysis of GhCAD6 During Cotton Fiber Development

To further investigate the expression specificity of *GhCAD6*, we selected positive single-copy transgenic progeny materials and analyzed the expression during five different days post-anthesis (DPA) of fiber development (5, 10, 15, 20 DPA). Compared to non-transgenic plants, RT-PCR analysis revealed that the expression of the *GhCAD6* gene increased with the progression of cotton fiber development in both transgenic plants and control plants (CK). Before 15 DPA, the expression of the *GhCAD6* gene in both transgenic progeny and controls was very low. However, at 20 DPA, the expression of the *GhCAD6* gene began to increase significantly. At 20 DPA, the expression of the *GhCAD6* gene was higher in transgenic progeny than in controls (Figure 2a). qRT-PCR analysis indicated that the expression of the *GhCAD6* gene in both transgenic progeny and controls was significantly higher at 20 DPA compared to 15 DPA. Additionally, the expression of the *GhCAD6* gene was higher in transgenic progeny than in controls at both 15 DPA and 20 DPA (Figure 2b). The results demonstrated that the *GhCAD6* gene in the progeny of transgenic materials was overexpressed at 15 and 20 DPA of fiber development, with the expression level at 20 DPA being higher than at 15 DPA.

### 2.3. GhCAD6 Regulates Lignin Synthesis in Cotton Fiber

Given that *GhCAD6* is predominantly expressed during the secondary wall synthesis stage in fiber cells (Figure 2), we employed the Klason method to detect lignin and lignin-like phenolic compounds in mature cotton fibers, aiming to determine whether *GhCAD6* regulates lignin biosynthesis. We measured the lignin content in both transgenic progeny and control plants of equivalent quality at various developmental stages of cotton fiber (20, 30, 40, 50 DPA, and mature). By observing the changes of cotton fiber samples at different development stages after the addition of 72% sulfuric acid for 1 h, it was found that the color of overexpression plants was darker (Figure 3a). The comparative map of lignin residue in cotton fiber on filter paper was observed. The results showed that the lignin content decreased with the increase in development days. Compared with the lignin content in transgenic progeny and control, it was found that the lignin content in transgenic progeny was higher than that in controls in the same development period (Figure 3b,c). These findings demonstrate that cotton fiber contains lignin, and the *GhCAD6* gene plays a positive regulatory role in lignin synthesis.

### 2.4. GhCAD6 Regulates the Synthesis of Cotton Fiber Cell Wall

The secondary cell wall constitutes the main component of mature cotton fibers. The stratification of the cell wall affects fiber quality formation. After observing the cell wall thickness of cotton fibers by electron microscopy, we found that the secondary wall deposition thickness of the OE-*GhCAD6* line was significantly thinner during fiber elongation (15–25 DPA) (Figure 4a,b). Notably, during the entire stage of fiber development, the cell walls of overexpression plants were thinner than those of the control plants (Figure 4a,b). Additionally, the surface of mature cotton fibers from transgenic progeny exhibited a smoother, finer, and tighter texture compared to the control fibers (Figure 4c). At the same time, we measured the length of mature fibers and found that the fiber lengths of OE-*GhCAD6* lines are longer (Figure 4d). We speculated that the *GhCAD6* gene can regulate the synthesis of the cotton fiber cell wall and increase the length of cotton fibers.

### 2.5. Observation of Agronomic Traits of Plants Overexpressing GhCAD6

During the growth of transgenic and control plants, observations were made of cotton bolls, seed cotton, plant height, and stem sections at different developmental stages (Figure 5). We found no change in plant height of OE-*GhCAD6* plants (Figure 5d). However, the transgenic cotton bolls were slightly larger than those of the control plants (CK) at different developmental stages (10, 15, 20, 25, 30, 40 DPA). At different developmental stages, the boll shape of the OE-*GhCAD6* line changed from round to oval (Figure 5a). After in vitro culture, the fiber length of OE-*GhCAD6* was significantly longer than that of the control (Figure 5b). At the same time, we also found that OE-*GhCAD6* plants were less infected with disease through the rod-cutting method, and it was speculated that the overexpression of *GhCAD6* may enhance cotton disease resistance (Figure 5c).

### 2.6. Analysis of Fiber Quality Parameters of GhCAD6-Overexpressing Plants

DPS software (Official Version 7.05) was used to analyze the data, and it was found that the fiber length, fiber uniformity, number of bolls per plant, specific strength at breakage, and lint percentage of the transgenic plants were higher than those of the control, among which the fiber length of the upper half of the transgenic plants increased by 17.1% (Figure 6a), the fiber uniformity increased by 2.7% (Figure 6b), the number of bolls per plant increased by 100% (Figure 6d), the specific strength at breakage increased by 24.2% (Figure 6c), and the lint percentage content increased by 7.2% (Figure 6f), and the micronaire value of transgenic plants decreased by 13.5% (Figure 6e). The results showed that overexpression of *GhCAD6* can improve the fiber quality in cotton.

## 3. Discussion

Several recent studies have suggested the potential synthesis of lignin or lignin-like phenols within cotton fiber cells [19]. Various classical chemical analysis methods, including Klason, mercaptoacetate, and acetyl bromide, support the presence of lignin-like phenolic compounds in cotton fiber. In this study, we generated 14 positive plants by constructing an overexpression vector and transferring it into cotton. Subsequently, we measured the lignin content in the expression plants using the Klason method and confirmed the presence of lignin and lignin-like phenols in cotton fiber.

### 3.1. GhCAD6 Participates in Lignin Synthesis

The *CAD* gene exhibited increased expression during the formation of secondary xylem in cotton fibers, playing a crucial role in lignin synthesis [34]. To verify the role of *CAD* genes in lignin synthesis, we selected the GhCAD6 gene from the CAD gene family based on previous studies. Through experiments such as transgenics, qRT-PCR, and lignin content determination, we discovered that *GhCAD6* was predominantly expressed during the secondary wall thickening stage of fiber development. Notably, the lignin content in the transgenic offspring was higher than that in the control group at the same stage of cotton fiber development, indicating that *GhCAD6* is involved in lignin biosynthesis, consistent with previous findings.

### 3.2. The Lignin Content Influences the Formation of Fiber Quality

It was found that the increase in lignin/lignin-like phenolic compound content resulted in a thinner fiber secondary wall, but the increase in density led to the formation of a tighter secondary wall in the cells and improved the fiber fineness and fiber strength of cotton fibers [35,36]. Next, we observed the changes in cell wall thickness and the surface characteristics of mature cotton fibers in transgenic progenies and control plants at different developmental stages. It was found that the cell walls of the transgenic progeny were thinner than those of the control plants. The surface of mature cotton fibers from the transgenic progeny was smoother, finer, and tighter compared to that of the control. According to previous research results, based on the functional role of lignin in cross-linked cellulose microfibrils, the increase in lignin/lignin-like phenolic substances in the fiber cells of transgenic plants may have enhanced the cohesion of cellulose microfibrils, even though the content of the latter remained unchanged [35]. At the same time, we also believe that this is precisely the reason for the formation of thinner and denser secondary cell walls. Furthermore, the lint percentage, fiber length, and specific strength at breakage of the transgenic offspring were higher than those of the controls, while the micronaire value decreased, ultimately leading to an improvement in the overall fiber quality of the transgenic offspring, consistent with previous studies [35]. However, the relevant mechanisms by which the *GhCAD6* gene regulates fiber quality formation still require further research.

### 3.3. An Increase in Lignin Content Will Increase Plant Disease Resistance

The cell wall serves as a barrier against pathogen infection [37], as it contains lignin and lignin phenols, which enhance plant resistance to pathogens. In this study, following the overexpression of *GhCAD6*, we observed an increase in lignin content in the plants, which exhibited a certain degree of disease resistance (Figure 5c). In the future, we will further verify whether changes in the expression level of *GhCAD6* can affect the disease resistance of cotton by combining phenotypic observation with data analysis.

## 4. Materials and Methods

### 4.1. Clone of GhCAD6 Gene and Construction of Overexpression Vector

In this study, we digested the plasmid pCAMBIA2300 with the restriction endonucleases *Sac* I and *BamH* I, cloned the CDS sequence of the *GhCAD6* gene (Appendix A) using the cottonFGD (https://cottonfgd.net/) (accessed on 15 October 2022) website, and ligated it into the overexpression vector pCAMBIA2300 by homologous recombination. After sequencing, we designated the construct as pCAMBIA2300-E6-*GhCAD6*. The gene editing vectors were then introduced into Agrobacterium tumefaciens strain GV3101 by electroporation and plated on Petri dishes containing kanamycin (Kan) and rifampicin (Rif) for selection. After monoclonal detection, the positive bacteria were preserved and stored at −80 °C.

### 4.2. Detection of Transgenic Plants

DNA was extracted from plant leaves using the CTAB method [38]. For PCR detection of transgenic offspring, the neomycin phosphotransferase gene (NPT II) present in the expression vector used in this experiment and specific primers capable of amplifying the *GhCAD6* gene were selected. The primers were designed using Primer Premier 5.0 software and synthesized by Beijing Liuhe Huada Genome Technology Co., Ltd. (Beijing, China). The template for PCR detection was the extracted genomic DNA.

### 4.3. Southern Blotting

The gel was immersed in a 10-fold volume of neutral denaturant solution for 45 min and oscillated slowly. A piece of nylon film was completely soaked in deionized water and subsequently placed in 20 × SSC solution for 5 min. The denatured DNA was transferred to the membrane by capillary action over a period of 24 h. Upon completion of the transfer, the membrane was washed with 2 × SSC and fixed at 120 °C for 30 min. The hybridization solution and membrane, in a volume 10 times that of the gel, were transferred to the hybridization chamber and pre-hybridized at 42 °C for 30 min. The labeled probe was boiled in a water bath for 5 min. Immediately after boiling, the probe was placed in an ice bath. The denatured probe was added to the hybridization solution at a ratio of 1:2500. The product and probe underwent agarose electrophoresis, membrane transfer, and hybridization. Subsequently, the membrane was placed in 50 mL of chromogenic solution, protected from light, and developed until a clear band appeared. The color reaction was stopped by washing the membrane with sterilized deionized water.

### 4.4. RNA Extraction and Expression Analysis of GhCAD6 Gene

Total RNA from cotton fibers at five developmental stages (5, 10, 15, 20 DPA) was extracted using an improved hot boric acid–protease K method. Based on the detected RNA concentration, the samples were diluted to approximately 0.25 μg/μL. A 1% agarose gel was prepared, and RNA concentration was verified via electrophoresis, followed by further RNA homogenization adjustments. Reverse transcription of sample RNA was conducted using TransScript One-Step gDNA Removal and cDNA Synthesis SuperMix (AT311-02) from Beijing Quanshijin Biotechnology Co., Ltd. (Beijing, China). RT-PCR was employed to compare the expression of the *GhCAD6* gene across different developmental stages in cotton samples (transgenic progeny lines and controls), while qRT-PCR was used to compare *GhCAD6* expression in cotton samples (transgenic progeny lines and controls) at the same developmental stage. Real-time quantitative PCR was performed using the fluorescent dye embedding method (SYBR Green I) and the 2^−ΔΔCt^ method [39]. GhEF1α was used as the internal reference [40].

### 4.5. Determination of Lignin Content in Plants

The lignin content in cotton fiber was determined using the Klason method [41]. Initially, an H-buffer was added to the sample, which was then boiled for 10 min. This process was repeated 2–3 times, followed by washing the sample twice with 2005 acetone and once with acetone. After each washing, the solution in the fiber was squeezed dry and the sample was finally dried and preserved. A quantitative filter paper was soaked in 3% sulfuric acid, washed to neutrality with deionized water, dried, and weighed, recording this weight as W1 (g). Two grams of the dried sample were placed into a tube, to which 15 mL of pre-cooled 72% sulfuric acid was added. The tube was kept until the cotton fiber was completely degraded, after which it was diluted with deionized water and sterilized at 121 °C for 1 h. After cooling, the filtrate was filtered and washed with deionized water until neutral. When tested with 10% Bacl_2_, no precipitation was observed in the filtrate. The filter paper was dried to a constant weight and recorded as W2 (g). The percentage content of lignin (%) was calculated using the formula: (W2 − W1)/2 × 100%.

### 4.6. Analysis of Cell Wall Structure and Fiber Properties

The cotton fiber samples were quickly put into a glass vial containing 4% glutaraldehyde (0.1 mol/L phosphate buffer with pH = 7.2) and capped and stored in a refrigerator at 4 °C for more than 5 h. The cotton fiber sample taken from the 4% glutaraldehyde solution was straightened on the filter paper, the middle part was carefully cut with a clean blade, it was laid flat in the embedding plate, and then the sample fiber was pre-embedded. Pre-embedding needs to be carried out in the embedding plate with 3.5% agar. When the fiber is completely solidified, the excess part outside the sample fiber is carefully cut off, leaving only the necessary fiber embedded, and finally it is put into a glass vial with a lid. The embedded fibers were washed with 0.1 mol/L phosphate buffer (pH = 7.2) for 3 h. The embedded fibers were fixed for 2 h with 1% osmic acid (0.1 mol/L phosphate buffer). Rinsing of the embedded fibers with 0.1 mol/L phosphate buffer continued for 1 h. The embedded fibers were gradually dehydrated with 30%, 50%, 70%, 80%, 90%, 95%, 100% ethanol, and each stage of ethanol dehydration lasted about 20 min, but 100% ethanol needed to be used for 1 h. The dehydrated fibers were soaked in anhydrous ethanol/acetone (1:1) for 1 h, anhydrous ethanol/acetone (1:4) for 1.2 h, acetone/epoxy resin (1:1) for 2 h, acetone/epoxy resin (1:3) for 3 h, acetone/epoxy resin (1:5) for 4 d, and epoxy resin for 1 d. The sample fibers were embedded in the embedded plate with epoxy resin and then placed in a constant temperature drying box at 37 °C, 45 °C, and 60 °C, respectively, for 1 day. The embedded sample fibers were sliced with a slicing machine. The slices were stained for 20 min in 2% uranyl acetate and 6% lead citrate, respectively, and washed with sterilized deionized water and dried. Finally, the slices were observed and photographed under an Hmur600 transmission electron microscope produced by Hitachi (Tokyo, Japan).

### 4.7. Analysis of Agronomic Traits of Plants

Four plants were randomly selected from transgenic progeny lines and control lines. Plant height and boll number per plant were measured, and the data were analyzed using DPS software.

The cotton lint percentage, fiber length, specific strength at breakage, and micronaire value are key indicators for determining fiber quality. Ten bolls from the transgenic progeny and ten bolls from the middle of the control plants were collected. The lint percentage was measured, and the obtained cotton fibers were sent to the Institute of Agricultural Quality Standards and Testing Technology at the Xinjiang Academy of Agricultural Sciences for quality assessment.

## 5. Conclusions

The *CAD6* gene is a gene related to lignin synthesis and is present in a variety of plants. Further research shows that *GhCAD6* positively regulates the synthesis of lignin and the development of fiber cells in upland cotton. The results of this study lay the foundation for elucidating the molecular mechanisms in which *GhCAD6* affects lignin synthesis and fiber quality formation.

## Figures and Tables

**Figure 1 ijms-26-09518-f001:**
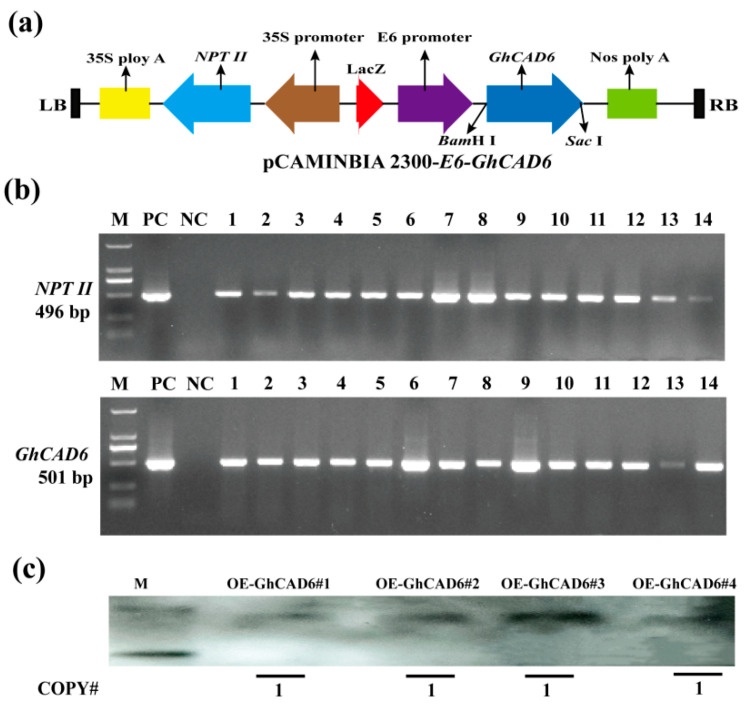
Acquisition and identification of *GhCAD6* overexpression plants. (**a**) Schematic diagram of pCAMINBIA 2300-E6-*GhCAD6* overexpression vector. (**b**) Positive detection of *GhCAD6* overexpression plants, PC: positive control; NC: negative control; 1–14: positive plants. (**c**) According to the copy number analysis of some overexpression plants.

**Figure 2 ijms-26-09518-f002:**
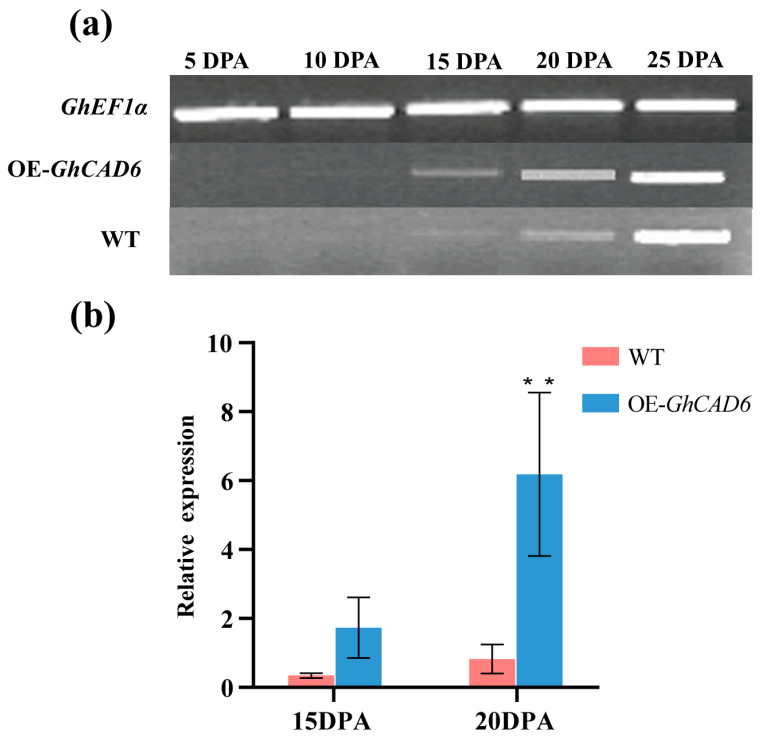
Expression analysis of *GhCAD6* gene during cotton fiber development. (**a**) RT-PCR analysis of *GhCAD6* gene in different developmental stages of cotton fiber. (**b**) The qRT-PCR analysis of *GhCAD6* gene in cotton fiber elongation. ** *p* < 0.01.

**Figure 3 ijms-26-09518-f003:**
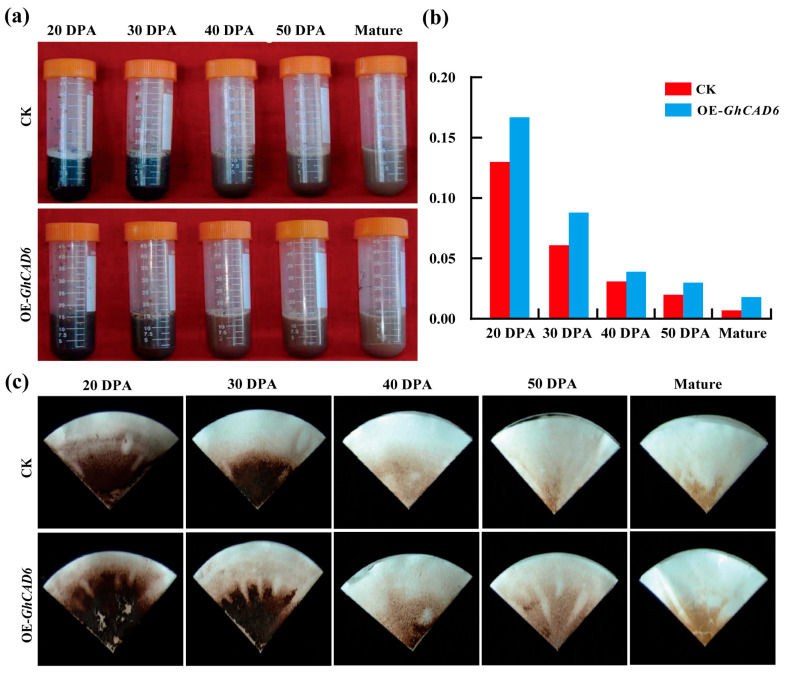
Determination of lignin content. (**a**) The change of 72% sulfuric acid after adding cotton fiber at different developmental stages for 1 h. (**b**) Lignin content in fibers of overexpressing plants and control plants. (**c**) Lignin residues in cotton fibers at different developmental stages on filter paper.

**Figure 4 ijms-26-09518-f004:**
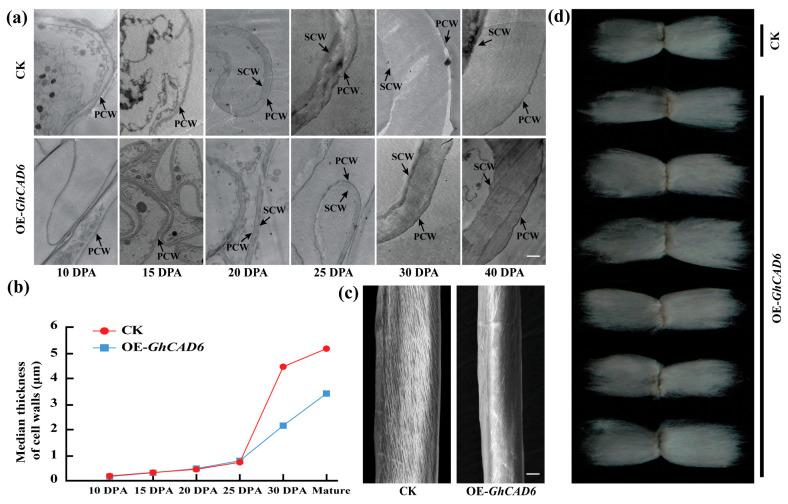
Observation of cotton fiber between *GhCAD6* overexpression plant and control. (**a**) Electron microscopic observation of cell wall thickness at different stages of cotton fiber development between *GhCAD6* overexpression plants and control, note. scw: Primary cell wall, pcw: Secondary cell wall, scale: 1 μm, filmed under the condition of ×10,000. (**b**) Comparison of cell wall thickness at different stages of cotton fiber development between *GhCAD6* overexpression plants and control. (**c**) Electron microscopic observation on the surface of mature cotton fiber of *GhCAD6* overexpression plant and control, scale: 1 μm, filmed under the condition of ×30,000. (**d**) Fiber images of overexpression plants and control plants.

**Figure 5 ijms-26-09518-f005:**
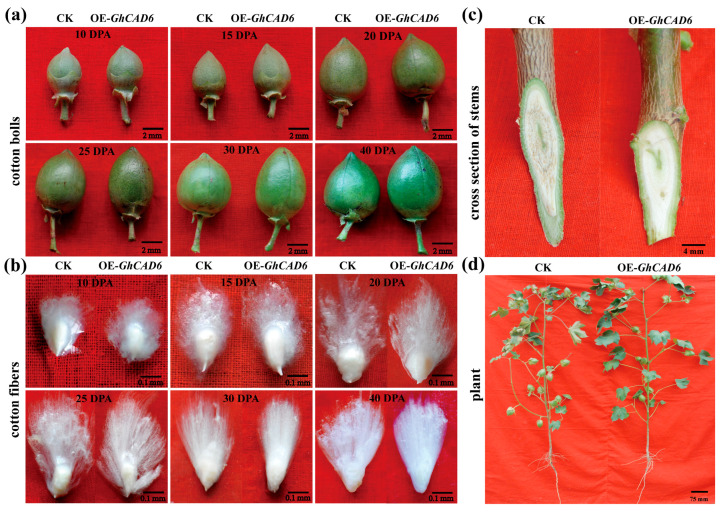
Observation of various characteristics of overexpression plants. (**a**) Observation of different developmental stages of cotton bolls from overexpression plants; scale: 2 mm. (**b**) Observation of seed cotton from overexpressing plants at varying developmental stages; scale: 0.1 mm. (**c**) Observation of the stem sections of overexpressing plants; scale: 4 mm. (**d**) Observation of the plant height of overexpressing plants; scale: 75 mm.

**Figure 6 ijms-26-09518-f006:**
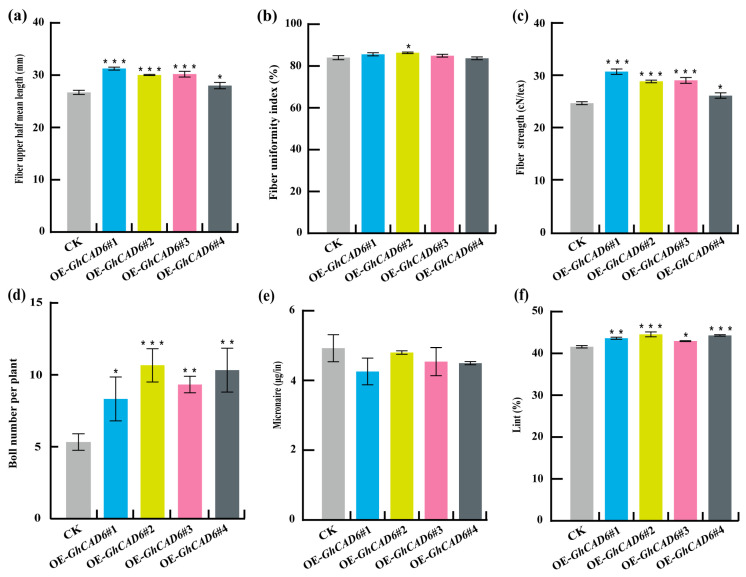
Comparison of fiber quality parameters between overexpression plants and control plants. (**a**) the fiber upper half mean length; (**b**) the fiber uniformity index; (**c**) the fiber strength; (**d**) the boll number per plant; (**e**) the micronaire values; (**f**) the lint percentage. * *p* < 0.05, ** *p* < 0.01, *** *p* < 0.001.

## Data Availability

The upland cotton variety ‘Zao36’ and the overexpression vector pCAMBIA2300 used in this study were kindly provided by the Xinjiang Uygur Autonomous Region Academy of Agricultural Sciences Biological Breeding Laboratory/Xinjiang Key Laboratory of Crop Biotechnology. The original contributions presented in this study are included in the article/Appendix A. Further inquiries can be directed to the corresponding authors.

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
