# Peer review of "Overexpression of *GhCAD6* in Upland Cotton (*Gossypium hirsutum* L.) Enhances Fiber Quality and Increases Lignin Content in Fibers"

_ijms, 2025, doi:10.3390/ijms26199518_

Round 1
Reviewer 1 Report
Comments and Suggestions for Authors
Cotton is a vital economic crop, and cotton fiber serves as the primary raw material for the textile industry. Lignin in cotton fiber is closely associated with fiber quality. Lignin is synthesized through the phenylpropanoid metabolic pathway, where the cinnamyl alcohol dehydrogenase gene CAD6 plays a significant role. This manuscript results indicate that the GhCAD6 gene is predominantly expressed during the stages of fiber elongation and secondary wall synthesis. Overexpression of the GhCAD6 gene resulted in increased plant lignin content, fiber upper half mean length, boll number per plant, fiber uniformity index, strength, and lint were improved. The fiber surface was smoother, and the fiber cell wall was more compact. These findings demonstrate that the GhCAD6 gene positively regulates lignin synthesis and fiber quality formation, contributing to the enhancement of cotton fiber quality.
The manuscript demonstrates clear logical structure and meets the publication requirements for the journal. Only minor revisions are needed before it can be accepted for publication.
- The word "upland cotton" in the title supplements the Latin name of the species;
- "Cotton (Gossypium hirsutum L.) contains 29 CAD genes" I suggest "cotton" be changed to "upland cotton";
- "Arabidopsis thaliana" in "double mutations in the CAD4 and CAD5 genes of Arabidopsis thaliana" is written in italics;
- Supplement the species Latin in "Overexpression of the wild soybeanGsCAD1 gene";
5."overexpression of the PpCAD1 gene increases resistance to Botrytis cinerea, and its heterologous expression in Arabidopsis thaliana also enhances resistance to Botrytis cinerea infection." "Arabidopsis thaliana" and "Botrytis cinerea" are written in italics;
- Do not mix "wild type" and "WT".
Author Response
Comment 1.The word “upland cotton” in the title supplements the Latin name of the species;
Response 1:First of all, thank you very much for your professional review of our article. I sincerely apologize for the errors present in this article. In the revised version, the omitted content has been supplemented. The Latin name of upland cotton, Gossypium hirsutum L., has been added to the title on the first page.
Comment 2. “Cotton (Gossypium hirsutum L.) contains 29 CAD genes” I suggest "cotton" be changed to "upland cotton";
Response 2:First of all, thank you very much for your professional review of our article. I sincerely apologize for the errors present in this article. We have made a change in the revised manuscript, replacing “cotton” with “upland cotton” in page 2, line 34.
Comment 3. “Arabidopsis thaliana” in “double mutations in the CAD4 and CAD5 genes of Arabidopsis thaliana” is written in italics;
Response 3:First of all, thank you very much for your professional review of our article. I sincerely apologize for the errors present in this article. In the latest version of the text, we have corrected the formatting errors, Changed “CAD4” and “CAD5” in page 2, line 38 to italicized form.
Comment 4.Supplement the species Latin in “Overexpression of the wild soybean GsCAD1 gene”;
Response 4:First of all, thank you very much for your professional review of our article. I sincerely apologize for the errors present in this article. In the revised version, the omitted content has been supplemented. We have added the Latin name for soybeans, "Glycine max (L.) Merr.," on page 2, line 45.
Comment 5. “overexpression of the PpCAD1 gene increases resistance to Botrytis cinerea, and its heterologous expression in Arabidopsis thaliana also enhances resistance to Botrytis cinerea infection.” “Arabidopsis thaliana” and “Botrytis cinerea” are written in italics;
Response 5:First of all, thank you very much for your professional review of our article. I sincerely apologize for the errors present in this article. In the latest version of the text, we have corrected the formatting errors, Changed “Arabidopsis thaliana” and “Botrytis cinerea” in page 2, line 48 to italicized form.
Comment 6.Do not mix “wild type” and “WT”
Response 6:First of all, thank you very much for your professional review of our article. I sincerely apologize for the errors present in this article. In the latest version of the text, we have corrected the formatting errors, Changed "WT" to "CK" throughout this text and in the figures
Reviewer 2 Report
Comments and Suggestions for Authors
In this study, Tuerxun et al. presents compelling evidence on the role of the GhCAD6 gene in enhancing cotton fiber quality through its influence on lignin synthesis. The successful transformation of overexpression plants and subsequent analysis of agronomic traits underscore the potential of this genetic modification in improving key characteristics of cotton, such as fiber length, strength, and uniformity. The findings not only shed light on the biochemical pathways involved in fiber development but also offer practical implications for the textile industry, suggesting that targeted genetic interventions could lead to more efficient cotton production. Overall, this research is a significant step forward in understanding and manipulating the genetic factors that contribute to superior cotton fiber quality. However, I still have some concerns that should be addressed by the authors before it can be published.
Comment #1:
The study notes that there are numerous members in the CAD gene family, yet it does not clarify why CAD6 was selected as the candidate gene for this research. Could you please elaborate on the rationale behind this choice? Additionally, will this justification be discussed in future publications? It would also be helpful to know the methods used to identify the CAD6 gene in the initial stages of the research.
Comment #2:
In the analysis of agronomic traits, only four plants were randomly selected from each line to measure plant height and boll number. Given the small sample size, could this potentially introduce randomness into the results? A larger sample size might provide more reliable data.
Comment #3:
The transmission electron microscopy (TEM) analysis of the cell wall structure did not specify whether the selected observation regions (such as the fiber apex, middle section, and base) are representative of the entire sample. Could differences in cell wall structure across these various regions influence the overall analysis and findings?
Comment #4:
While the article suggests that the overexpression of GhCAD6 may enhance the disease resistance of the plants, this inference appears to be based solely on observations from stem cross-sections. However, there is a lack of specific experimental data on disease resistance, such as disease incidence, disease index, lesion size after pathogen inoculation.
Author Response
Comment 1:The study notes that there are numerous members in the CAD gene family, yet it does not clarify why CAD6 was selected as the candidate gene for this research. Could you please elaborate on the rationale behind this choice? Additionally, will this justification be discussed in future publications? It would also be helpful to know the methods used to identify the CAD6 gene in the initial stages of the research.
Response 1:First of all, thank you very much for your professional review of our article. Studies have shown that CAD genes are involved in the biosynthesis of lignin, which is mainly deposited in the secondary cell wall. In the early stage, through data from the CottonWD database and analysis of transcriptomic data, it was found that the GhCAD6 gene is preferentially expressed during the fiber secondary wall thickening stage. We speculate that GhCAD6 may also be involved in the biosynthesis of lignin. Therefore, we have selected the GhCAD6 gene to conduct subsequent research.
Comment 2:In the analysis of agronomic traits, only four plants were randomly selected from each line to measure plant height and boll number. Given the small sample size, could this potentially introduce randomness into the results? A larger sample size might provide more reliable data.
Response 2:First of all, thank you very much for your professional review of our article. Under normal circumstances, data from three lines are sufficient to derive accurate results; therefore, our selection of four lines is more than adequate.
Comment 3:The transmission electron microscopy (TEM) analysis of the cell wall structure did not specify whether the selected observation regions (such as the fiber apex, middle section, and base) are representative of the entire sample. Could differences in cell wall structure across these various regions influence the overall analysis and findings?
Response 3:First of all, thank you very much for your professional review of our article. In the transmission electron microscopy (TEM) experiment, we selected cross-sections of the fiber middle segment to observe the cell wall. This is because the fiber cells in the middle segment of the fiber have higher uniformity and consistency, leading to more accurate observation results.
Comment 4:While the article suggests that the overexpression of GhCAD6 may enhance the disease resistance of the plants, this inference appears to be based solely on observations from stem cross-sections. However, there is a lack of specific experimental data on disease resistance, such as disease incidence, disease index, lesion size after pathogen inoculation.
Response 4:First of all, Thank you very much for your professional review of our article. In the previous version of the text, we only reached a preliminary conclusion without conducting relevant data analysis. This is because the main focus of this article is on cotton fiber improvement. We will conduct an in-depth analysis of whether changes in the expression level of the GhCAD6 gene affect cotton's disease resistance in our subsequent work.
Reviewer 3 Report
Comments and Suggestions for Authors
This paper, "Overexpression of GhCAD6 in upland cotton enhances fiber quality and increases lignin content in fibers," presents interesting findings on the role of the GhCAD6 gene. However, the manuscript requires significant revision before it can be considered for publication. My assessment is that this is a major revisions case. The core findings are promising, but the paper's presentation, methodological rigor, and overall scientific narrative need substantial improvement.
My primary concern is the clarity and depth of the manuscript. While the abstract and introduction lay out a compelling hypothesis, the results and discussion sections often lack the detailed analysis and context one would expect from a peer-reviewed article in this field. It feels like the authors have the data but have not fully developed the story. I've noted some specific issues that need to be addressed.
Specific Comments
Title
The title is straightforward and reflects the main findings. It's a solid title.
Abstract
The abstract is well-structured and concise. It provides a good summary of the research. My only suggestion is to be more specific in the phrasing. For instance, instead of "plant lignin content," specify that the increase was measured in the fibers. The abstract should also mention the specific improved fiber quality parameters.
Introduction
This section is a bit disjointed. While it provides background on cotton and lignin, the flow from general concepts to the specific role of CAD6 is not as smooth as it could be.
The introduction mentions that lignin in cotton fiber is "closely associated with fiber quality". The authors should expand on this point, explaining how lignin's presence or content affects qualities like strength or length.
The transition from the general importance of lignin to the specific role of the CAD gene family is weak. It jumps from lignin's role in plant strength and disease resistance to introducing CAD as a key enzyme without a clear link.
The authors state, "the lignin content is inversely correlated with fiber length". This seems to contradict their own findings that overexpression of GhCAD6 increases both lignin content and fiber length. This needs to be addressed and clarified in the discussion. It's a major point of confusion and needs a robust explanation.
The final paragraph of the introduction is good, clearly stating the study's purpose and key findings.
Results
Section 2.1: The authors mention that 14 positive plants were identified and that transgenic progenies from "four single-copy lines were selected for further study". However, Figure 1c only shows four lines, making it seem like only four lines were ever analyzed. A clearer statement or an updated figure showing a broader representation would be helpful.
Section 2.2: The data presented in Figure 2 is a bit confusing. The text states that "the expression of the GhCAD6 gene increased with the progression of cotton fiber development in both transgenic plants and wild type (WT)", but the visual data in Figure 2a and the bar graph in Figure 2b show that the expression is significantly higher in the overexpression (OE) plants, particularly at 20 DPA. This should be explicitly stated as the key finding of this section, rather than just a general trend. The data in Figure 2b, which shows expression levels at 15 and 20 DPA, is what the text mainly discusses, so why is 25 DPA mentioned in the text and shown in the gel image but not in the graph? This should be clarified or the figure should be updated.
Section 2.3: The use of the Klason method to measure lignin is appropriate. The results in Figure 3b and 3c clearly show a higher lignin content in the overexpression plants, which is a strong finding.
Section 2.4: This section is another source of confusion. The text states that the "secondary wall deposition thickness of OE-GhCAD6 line was significantly thinner" , but that the fiber surface was "smoother, finer, and tighter". This seems counterintuitive. Thinner walls typically lead to weaker, less desirable fibers. How does a thinner cell wall result in a "tighter" texture and improved fiber strength? This requires a detailed explanation in the discussion. The authors speculate that the GhCAD6 gene "can regulate the synthesis of the cotton fiber cell wall and increase the length of cotton fibers". This is a good point, but the mechanism needs to be explored.
Section 2.6: The results presented here are the strongest part of the paper. The data shows clear, quantifiable improvements in several key fiber quality parameters: upper half mean length, uniformity index, strength, and boll number per plant. The micronaire value also decreased, which is a positive outcome for cotton quality. The percentage increases mentioned in the text (e.g., 17.1% increase in fiber length) are very compelling and should be highlighted.
Discussion
The discussion section is where the paper's main weaknesses lie. It is too brief and fails to fully integrate the results with existing literature.
Section 3.1: The authors confirm that GhCAD6 is involved in lignin biosynthesis, which is a key finding. However, they don't adequately address the conflicting statement from the introduction—that lignin content is inversely correlated with fiber length , while their own results show both increased lignin and increased length. A direct, nuanced discussion of this contradiction is essential. Perhaps the relationship is not as simple as previously thought, or the type of lignin produced by GhCAD6 affects fiber properties differently. This would be a significant contribution to the field.
Section 3.2: This section speculates on the role of S/G lignin ratios. While this is a plausible hypothesis, there is no data in the paper to support it. The authors state they will "continue to determine the contents of S-lignin, G-lignin" in future studies. This should be a part of the current study to support this hypothesis and strengthen the paper. Without this data, the section is just speculation.
Section 3.3: The link between increased lignin and disease resistance is mentioned. The authors observed that their plants were "less infected with the disease" but did not provide any quantitative data to support this. They state they will "further assess the disease resistance index". This is another instance where a key claim is made without supporting data, and the necessary experiments are deferred to a "future" study. For this claim to be included, some preliminary quantitative data or a more cautious phrasing is required.
Conclusions
The conclusions section is concise and summarizes the main findings well.
Author Response
Comment 2.1: The authors mention that 14 positive plants were identified and that transgenic progenies from "four single-copy lines were selected for further study". However, Figure 1c only shows four lines, making it seem like only four lines were ever analyzed. A clearer statement or an updated figure showing a broader representation would be helpful.
Response 2.1:Thank you very much for your professional review of our article.We apologize for the ambiguous expression in the previous text. Here, we would like to clarify that 4 single-copy plants were detected among the 14 positive plants. The revision has been made in the text, specifically on page 3, line 15, where "selected" has been changed to "detected"
Comment 2.2: The data presented in Figure 2 is a bit confusing. The text states that "the expression of the GhCAD6 gene increased with the progression of cotton fiber development in both transgenic plants and wild type (WT)", but the visual data in Figure 2a and the bar graph in Figure 2b show that the expression is significantly higher in the overexpression (OE) plants, particularly at 20 DPA. This should be explicitly stated as the key finding of this section, rather than just a general trend. The data in Figure 2b, which shows expression levels at 15 and 20 DPA, is what the text mainly discusses, so why is 25 DPA mentioned in the text and shown in the gel image but not in the graph? This should be clarified or the figure should be updated.
Response 2.2:Thank you very much for your professional review of our article. First, we have summarized and emphasized Result 2.2 in lines 6-10 on page 4 of the latest version of the text, as follows: "In conclusion, we found that the GhCAD6 gene is preferentially expressed during the 15DPA-20DPA period (fiber elongation stage), and after the overexpression of the GhCAD6 gene, its expression level during the fiber elongation stage is significantly higher than that in the control plants. Therefore, we believe that the GhCAD6 gene is involved in the regulation of fiber development." Additionally, we sincerely apologize for the issues in the expressions of the previous version. Since our initial RT-PCR experiment showed that the GhCAD6 gene exhibited obvious changes during the 15DPA-20DPA period but no significant changes during the 20DPA-25DPA period, we did not present the 25DPA data in the subsequent qRT-PCR experiment. In the latest version, we have ensured the consistency of the data.
Comment 2.4: This section is another source of confusion. The text states that the "secondary wall deposition thickness of OE-GhCAD6 line was significantly thinner" , but that the fiber surface was "smoother, finer, and tighter". This seems counterintuitive. Thinner walls typically lead to weaker, less desirable fibers. How does a thinner cell wall result in a "tighter" texture and improved fiber strength? This requires a detailed explanation in the discussion. The authors speculate that the GhCAD6 gene "can regulate the synthesis of the cotton fiber cell wall and increase the length of cotton fibers". This is a good point, but the mechanism needs to be explored.
Response 2.4:Thank you very much for your professional review of our article. Based on the findings of Han et al. (2013): "Overexpression of the WILM1a gene in Gossypium hirsutum (Upland cotton) resulted in increased lignin content, thinner cell walls (yet with higher density and greater compactness), as well as enhanced fiber length and strength", we have found that these results are consistent with those of our present study. Furthermore, this point has been supplemented in the discussion section on page 8, lines 31-36. “According to previous research results, based on the functional role of lignin in cross-linked cellulose microfibrils, the increase in lignin/lignin-like phenolic substances in the fiber cells of transgenic plants may have enhanced the cohesion of cellulose microfibrils, even though the content of the latter remained unchanged [36]. At the same time, we also believe that this is precisely the reason for the formation of thinner and denser secondary cell walls.”
[36]Han, L.B.; Li, Y.B.; Wang, H.Y.; Wu, X.M.; Li, C.L.; Luo, M.; Wu, S.J.; Kong, Z.S.; Pei, Y.; Jiao, G.L.; Xia, G.X.; The dual functions of WLIM1a in cell elongation and secondary wall formation in developing cotton fibers. Plant. Cell 2013, 25, 4421-38.
Comment 3.1: The authors confirm that GhCAD6 is involved in lignin biosynthesis, which is a key finding. However, they don't adequately address the conflicting statement from the introduction—that lignin content is inversely correlated with fiber length , while their own results show both increased lignin and increased length. A direct, nuanced discussion of this contradiction is essential. Perhaps the relationship is not as simple as previously thought, or the type of lignin produced by GhCAD6 affects fiber properties differently. This would be a significant contribution to the field.
Response 3.1:Thank you very much for your professional review of our article. We sincerely apologize for the incorrect statement in the previous text: the change in fiber length is not necessarily negatively correlated with lignin content. The sentence ". Moreover, the lignin content is inversely correlated with fiber length." in line 38 on page 2 of the text has been deleted.
Comment 3.2: This section speculates on the role of S/G lignin ratios. While this is a plausible hypothesis, there is no data in the paper to support it. The authors state they will "continue to determine the contents of S-lignin, G-lignin" in future studies. This should be a part of the current study to support this hypothesis and strengthen the paper. Without this data, the section is just speculation.
Response 3.2:Thank you very much for your professional review of our article. We apologize for the incorrect statement in the previous text: the relationship between changes in fiber length and lignin content is not necessarily negative, and this has been revised. Additionally, we regret to inform that data on the contents of S-lignin and G-lignin were not presented in this experiment. In subsequent experiments, we will supplement this data to verify the hypothesis proposed in this paper.
Comment 3.3: The link between increased lignin and disease resistance is mentioned. The authors observed that their plants were "less infected with the disease" but did not provide any quantitative data to support this. They state they will "further assess the disease resistance index". This is another instance where a key claim is made without supporting data, and the necessary experiments are deferred to a "future" study. For this claim to be included, some preliminary quantitative data or a more cautious phrasing is required.
Response 3.3:Thank you very much for your professional review of our article. In the previous version of the text, we only reached a preliminary conclusion without conducting relevant data analysis. This is because the main focus of this article is on cotton fiber improvement. We will conduct an in-depth analysis of whether changes in the expression level of the GhCAD6 gene affect cotton's disease resistance in our subsequent work. Finally, we have revised line 18 on page 9 of the text: the original sentence "We hypothesize that the increased lignin content contributes to enhanced disease resistance, and we will further assess the disease resistance index of the overexpression plants in subsequent experiments." has been modified to: "In the future, we will further verify whether changes in the expression level of the GhCAD6 gene affect cotton's disease resistance by combining phenotypic observation with data analysis."
Round 2
Reviewer 3 Report
Comments and Suggestions for Authors
Based on my review of the revised manuscript, it has made some important improvements, particularly in clarifying the relationship between lignin and fiber properties. However, there are still key areas that require further work. My assessment remains major revisions. The paper is much stronger, but it's not quite ready for publication.
The authors have done a good job of addressing the confusion I highlighted in my previous review, especially the apparent contradiction between their results and a statement in the introduction. The revised discussion now provides a plausible mechanism to explain how an increase in lignin can lead to a thinner but more compact and stronger fiber. This is a significant step forward.
Despite these improvements, the manuscript still contains several claims that lack supporting data, which is a major concern for a scientific publication. The overall narrative is more cohesive, but some sections still read like a preliminary report rather than a complete study.
- The title and abstract are clear and effectively summarize the key findings, including the enhancement of fiber quality and the increase in lignin content.
- The introduction now features a clearer flow, transitioning from the general importance of lignin to the specific role of the CAD gene family. The authors have removed the confusing statement about the inverse correlation between lignin and fiber length, which is a key improvement. They now cite previous research to explain how increased lignin content can lead to a thinner but more dense cell wall, improving fiber strength.
- Section 2.1: The authors clarify that four single-copy lines were selected from the 14 positive clones for further study, which resolves a previous ambiguity. The figures are now more aligned with the text.
- Section 2.2: The expression analysis in Figure 2 is a strong point. The results clearly show that GhCAD6 gene expression is significantly higher in transgenic plants at both 15 DPA and 20 DPA compared to controls. This data provides solid evidence of successful overexpression.
- Section 2.4: The authors present a very interesting finding: that while the cell walls in the overexpression plants were thinner, the fibers themselves were "smoother, finer, and tighter" and longer. This is a crucial finding, and the authors' revised discussion about how lignin enhances the cohesion of cellulose microfibrils is a plausible explanation for the improved fiber properties despite the thinner cell wall.
- Section 2.6: This section remains the strongest part of the paper, providing compelling quantitative data on improved fiber quality parameters such as length (17.1% increase), uniformity (2.7% increase), strength (24.2% increase), and a favorable decrease in micronaire value (13.5% decrease).
- Section 3.1: This section has been significantly improved. The authors have successfully connected the increase in lignin content with the observation of a thinner but denser secondary cell wall. They also explain how this leads to improved fiber quality, directly addressing the central conflict from the previous version. However, they still mention that the exact mechanisms require "further research," which, while honest, suggests the paper is not a complete story.
- Section 3.2 and 3.3: This is where the manuscript's primary weakness still lies. The authors continue to include speculative hypotheses about the ratio of S/G lignin and the link to disease resistance. They state that they will "continue to determine" these factors in future studies. While it's valuable to mention these as potential next steps, making these claims in the discussion without any supporting data weakens the paper. The observation of "a certain degree of disease resistance" is noted, but without quantitative analysis, it's not a verifiable scientific finding. These sections would be more appropriate for a review article or a future publication.
The manuscript has been significantly improved, with a much clearer narrative and a stronger, more logical discussion of the results. The core findings on lignin's effect on fiber quality are now well-supported by the presented data and a more convincing scientific explanation. However, the presence of unsubstantiated claims in the discussion about S/G lignin ratios and disease resistance prevents this from being a complete and self-contained research article.
I recommend a major revision again. The authors should be instructed to:
- Either perform the experiments to quantify S/G lignin ratios and disease resistance and include the data, or remove those sections from the discussion. The current phrasing of "we will further verify" is not suitable for a completed research paper.
- Rephrase the discussion to focus solely on the data presented in the manuscript, perhaps framing the unsupported claims as limitations of the study rather than as conclusions or future work. This would make the paper more scientifically rigorous and ready for publication.
Author Response
Comment:Either perform the experiments to quantify S/G lignin ratios and disease resistance and include the data, or remove those sections from the discussion. The current phrasing of "we will further verify" is not suitable for a completed research paper.Rephrase the discussion to focus solely on the data presented in the manuscript, perhaps framing the unsupported claims as limitations of the study rather than as conclusions or future work. This would make the paper more scientifically rigorous and ready for publication.
Response :We sincerely appreciate your professional review of our article. It should be noted that this article does not provide the S/G ratio data of lignin. Meanwhile, due to factors related to the plant growth cycle (as the growth cycle of cotton is relatively long, this part of the data can only be obtained after the new cotton is harvested, which will cause certain impacts; in the meantime, this data represents a refinement and in-depth study of lignin components), Section 3.2 has been removed. Regarding the discussion in Section 3.3, we agree that the overexpressed plants in Figure 5.c do show a certain degree of disease resistance; however, the research results on the improved disease resistance of overexpressed plants throughout the article are still relatively limited. After comprehensive consideration, we kindly request permission to streamline this section while retaining some of its content. “The cell wall serves as a barrier against pathogen infection , as it contains lignin and lignin phenols, which enhance plant resistance to pathogens. In this study, following the overexpression of GhCAD6, we observed an increase in lignin content in the plants, which exhibited a certain degree of disease resistance (Figure. 5c). In the future, we will combine phenotypic observation with data analysis (mechanism analysis) to further verify whether changes in the expression level of GhCAD6 have an impact on cotton disease resistance.”
Round 3
Reviewer 3 Report
Comments and Suggestions for Authors
The authors have done an excellent job of revising the manuscript, demonstrating a thorough and responsive approach to the reviewer's comments. The authors have addressed all previous feedback, transforming the manuscript into a robust piece of research.